# Teachers Supporting Students in Collaborative Ways—An Analysis of Collaborative Work Creating Supportive Learning Environments for Every Student in a School: Cases from Austria, Finland, Lithuania, and Poland

Suvi Lakkala [1,*], Alvyra Galkienė [2], Julita Navaitienė [2], Tamara Cierpiałowska [3], Susanne Tomecek [4] and Satu Uusiautti [1]

1 Faculty of Education, University of Lapland, 96300 Rovaniemi, Finland; satu.uusiautti@ulapland.fi
2 Educational Academy, Vytautas Magnus University, 44248 Kaunas, Lithuania; alvyra.galkiene@vdu.lt (A.G.); julita.navaitiene@vdu.lt (J.N.)
3 Institute of Special Education, Pedagogical University of Krakow, 30-060 Krakow, Poland; tamara.cierpialowska@up.krakow.pl
4 Institut für übergreifende Bildungsschwerpunkte, University College of Teacher Education Vienna, 1100 Vienna, Austria; susanne.tomecek@phwien.ac.at
* Correspondence: suvi.lakkala@ulapland.fi

**Abstract:** Many studies have highlighted the importance of community and cooperation in inclusive education. However, traditionally, teachers are trained to manage their classes alone. Along with the aspirations of inclusive education, there is high pressure to develop school cultures that are more communal and to reorient school personnel's work, making it more collaborative, in order to meet the diverse needs of all students. In this research, we explored and compared the collaborative ways in which teachers supported their students in four schools in Austria, Finland, Lithuania, and Poland. As a conceptual framework, the research utilized theories of interprofessional teamwork. The researchers applied a theory-led thematic analysis to the research data. The main findings indicate that collaborative action needs to be an essential part of teachers' work in an inclusive school. The schools and teachers implemented both proactive and reactive ways of constructing an inclusive pedagogy when they supported their students.

**Keywords:** collaboration; co-teaching; inclusive education; inclusive pedagogy; interprofessional teamwork

## 1. Introduction

During the last few decades, pedagogies and teaching have evolved to be more student-oriented and flexible in their methods [1]. This is due to the internationally approved goal of inclusive education, expressed in many international guidelines, for example in a Guide for Ensuring Inclusion and Equity in Education by UNESCO [2,3]. According to Slee [4], the goal of inclusive education is to remove learning barriers, involving students and enabling them to learn in their own learning community. The move towards inclusive education has created pressure to change the whole school culture, along with teachers' professional identity and role [5]. Inclusive education is based on communality and collaboration among professionals as well as students and parents [6]. As such, communality in inclusive education refers to social theories [7]. However, "there is a need for greater understanding about ways in which individuals develop commitment to the processes of joint work, and how such motivation can resolve inter-professional dilemmas" [8], pp. 59–60.

For centuries, teachers have been trained to work alone and to have control over their classes [9]. The traditional strategy of managing a class on one's own has slowed down the successful implementation of inclusive education [10]. Many studies have found that teachers feel they do not have sufficient competence to teach diverse students [11,12].

One reason may be that there are only a few systematic pedagogical descriptions of how to implement inclusive education [5]. Another reason may be the difficulty of teachers changing their professional identity from a lone worker to a person who is able to work and learn together with others in a multiprofessional network [13].

The practical implementation of inclusive education varies greatly in different European countries in terms of the interpretation of concepts as well as implementation strategies for inclusive education and their quality [14]. More research is needed, not only to enhance common understanding about the basic principles of inclusive education, but also to find the best practices and solutions to implement inclusion. Previous research has found a strong connection between students' feeling of belonging and learning achievements [15]. For example, a supportive school climate and safe teacher-student relationships increase students' engagement [16,17]. Many studies state that students' sense of community and the satisfaction of their educational needs increase students' engagement with their studies [18]. However, educational support is difficult to investigate because of its variability. Research still lacks empirical studies that investigate and analyze educational support in various educational environments in different countries. More and more, scholars have become aware that a teacher working alone inside the classroom is not able to pay attention to the learning of every student because their needs are very diverse [10]. Instead, teachers' will and ability to continuously reflect on their professional actions with colleagues and other professionals have been found to be crucial [5,19].

Changes in educational policies call for schools to structure their professional work in new ways. The drive towards inclusive education necessitates a closer collaboration with other professions and sectors in order to promote educational support for all students in inclusive settings. That is why, in this research, we tie the building of educational support to the research discussions about interprofessional and collaborative networks [20,21]. The importance of promoting support by interprofessional teamwork has been outlined in the literature, but only few studies have described interprofessional practices involving teachers [22].

By interprofessional teamwork, we mean goal-oriented cooperation between different professions in which new kinds of knowledge and operating models emerge [23]. It contains the idea of looking at something together from the perspectives of one's own and another's profession, learning from professionals in other fields [8]. The difficulty of interprofessional teamwork lies in the dilemmas that may arise from the different professional orientations and each team member's abilities to detect his/her own role as a promoter of collaboration and negotiator of new goals in boundary spaces [8,24]. The premise of interprofessional teamwork is shared expertise, whereby team members create goals and strategies together, instead of offering ready-made solutions based on one's own profession [23].

In this study, researchers from four European countries—Austria, Finland, Lithuania, and Poland—compared the collaborative ways in which teachers in each country developed supportive learning environments for every student in their work. In this study we concentrate especially on the teachers' professional network. The research utilizes the results of our previous ethnographic research executed in each country's unique context [25]. By exploring one school case in each country, we examined how teachers constructed their teaching and support in trying to prevent failures in learning for their students and promoting inclusive education. The research was carried out in 2014–2017, within a European research program called Inclusive Education: Social-psychological, educational, and social aspects (Erasmus +, KA2). For this article, we performed a comparative analysis of the collaborative dimensions needed for creating supportive learning environments for all students.

## 2. Theoretical Background

### 2.1. The Collaborative Professional Orientation of an Inclusive Teacher

The implementation of inclusive education demands dedication and determination from teachers. For example, in the Profile of Inclusive Teachers (PIT), inclusive teachers'

areas of competence are listed as attitudes and beliefs, knowledge, and skills. The areas of competence are expressed on the basis of four core values, describing the "values in action" [14]. The four core values linked to the areas of competence are presented in Table 1.

**Table 1.** Four core values underpinning areas of competence for inclusive education [26], p. 199.

| Core Value | Necessary Areas of Competence Related to |
|---|---|
| 1. Valuing student diversity<br>-student difference is considered as a resource and asset to education | Conceptions of inclusive education<br>The teacher's view of learner difference |
| 2. Supporting all learners<br>-teachers have high expectations for all learners' achievement | Effective teaching approaches in heterogeneous classes<br>Promoting the academic and social learning of all learners |
| 3. Working with others<br>-collaboration and teamwork are essential approaches for all teachers | Working with parents and families<br>Working with a range of other educational professionals |
| 4. Continuing professional development<br>-teaching is a learning activity and teachers must accept responsibility for their own lifelong learning | Teachers as reflective practitioners<br>Initial teacher education as a foundation for ongoing professional learning and development |

The PIT has similarities with Florian and Spratt's [5] analytical framework they created in research developing principles for a one-year Professional Graduate Diploma in Education (PGDE), applying inclusive education. According to the first principle of the PGDE, an inclusive teacher must give up deterministic views of ability and see students as transformative individuals. The second principle states that the difficulties pupils experience in learning can be reflected as dilemmas for teaching rather than problems with the pupils themselves. The third principle claims that the profession must continuously develop new and creative ways of working with others. Both the PIT and the PGDE framework capture the core element of inclusive education: the inclusive school responds to the needs of its students, instead of sending the students to various environments that are considered to best respond to their learning needs. Inclusive education is meant for all students, not just those who have special needs [5,14].

Both previously introduced frameworks for inclusive teachers' orientation or competencies entail the dimension of collaboration and working with others. It is commonly acknowledged that the strength of inclusive teaching lies in its diversity on multiple levels, not just being addressed to certain groups of students, like special needs students [27]. For example, Tjernberg and Mattson [19] stress the importance of keeping an eye on students' learning process and finding the individual strengths of each student through inclusive teaching. Their notion of teachers' constant reflectivity and care for all of their students has strong implications for student-oriented teaching, which addresses pedagogical approaches that facilitate students' participation and possibilities to learn and progress, whatever abilities the students may have at any given moment [1]. The teaching in heterogeneous groups naturally leads to a need for carrying out various teaching methods that are implemented according to the students' needs, i.e., differentiated teaching [28]. In turn, differentiation presumes resources, such as time, materials, physical classroom space, and personnel [12,29].

However, along with the aspirations towards inclusive education, the school has become more aware of the complex problems that arise in heterogeneous classes. That is why collaboration (for example, team teaching) has become more and more common in schools [30]. Team teaching can be considered one form of interprofessional teamwork. Team teaching means a collaboration between two or more teachers, often a classroom teacher and a special education teacher, where they are responsible for planning, implementing, and assessing the teaching of a group of diverse students [31]. In order to create a solid professional relationship among the collaborators, it is necessary to reflect each member's personal views and expectations, have mutual communication and reflective discussions in the team, as well as be flexible and listen to other members' opinions [23].

The practical challenges related to teamwork are, for example, a paucity of mutual planning time or scheduling team teaching [30]. More profound prerequisites arise from the

professional relationship with the partners [8]. Difficulties may appear, for example, due to team members' commitment to different goals, unequal positions between the teachers, or disagreeing about the other team members' teaching philosophy or actions [32].

In inclusive education, teachers often collaborate with other professionals. For example, in all of our school cases there were several professionals working with the students, their families, and teachers during the school day. When the collaborative network expands beyond the teaching staff, there are even more possibilities for misunderstandings between parties who represent different professional backgrounds [24]. In interprofessional teamwork, the role of an individual as a part of the group [8] and the social and professional relationships in the team [23] are important. The concept of "relational agency" [7] refers to the extended learning that may develop in interprofessional teamwork and addresses the focus on the collaborative processes, specifically the capacity of individuals to develop shared understanding in joint work. Relational agency refers to the boundary areas between different professionals, where there are collective goals for the team but also the need to combine team members' various competences [8]. Success in interprofessional teamwork gives rise to one's own professional development and a multidisciplinary understanding when team members learn new perspectives from other professionals [7].

Rose and Norwich [8] were interested in the processes that lead to team members' commitment to collective goals or, alternatively, may cause dilemmas and tensions in teamwork. Based on a conceptual analysis of the literature, they developed theoretical frameworks of interprofessional teamwork. They used both social-level analysis and social-psychological-level analysis. The social-level analysis consists of factors affecting interprofessional teamwork, such as national and local government policies and structures, as well as the regulations and codes of practice of different services and professions. During collaboration, these regulations interact, and tensions between the different policies may appear, which, in turn, affect to the group's work. Concerning the social-psychological-level analysis, the researchers came up with a description of the positive collective preferences, referring to the group and the individual team members, as follows:

(1)    the group prefers and intends to achieve the best outcome for the group;
(2)    the individual acts as a part of the group to achieve this outcome;
(3)    the individual is accountable to the rest of the group for their actions (p. 66).

The framework is based on concepts of collective commitment, collective efficacy, and process and outcome beliefs. In addition, Rose and Norwich [8] highlighted the tensions that may arise in collaborative work. They are linked to team members' dilemmas in terms of their professional role, identity, and control in multi-agency work.

### 2.2. A Glance at Education in Austria, Finland, Lithuania, and Poland

This research focuses on four school cases implementing inclusive education in four European countries: Austria, Finland, Lithuania, and Poland. The participating countries' diverse political and social background affects their educational cultures. Two of our participating countries, Finland and Austria, have a long history of democratic development. The Finnish representative democracy has been in place since 1917 [33]. The independent Austrian state, restored after World War II, marked the beginning of its democratic development in 1955 [34]. In turn, the other two countries, Lithuania and Poland, have gone through drastic social unification processes during the years of Soviet occupation, and have experienced active revolutionary movements while liberating themselves from the regime. In 1989, Poland was named the Democratic Republic of Poland, and its development into a democratic state begun [35]. Lithuanian people escaped the Soviets' grip when they brought about the so-called "Singing Revolution," and when the Act of the Supreme Council of Lithuania on the re-establishment of the independent state of Lithuania was adopted on 11 March 1990 [36]. Political and social changes in social structures have a direct impact on the development of educational systems [37]. Although the educational systems in the four countries vary, the domination of strong state schools is one of the main features of all of them.

Austria has a differentiated school system. At the primary level, pupils are in "integrative/inclusion classes" taught either by one regular and one special schoolteacher, or supported by special education teachers for a limited number of hours per week. The special schoolteachers are formal members of a center for special education support, although they may work full-time at schools. Pupils with special educational needs (SEN) may also attend special schools. After the fourth grade, the pupils take an exam and may continue in three parallel pathways: general secondary school, academic secondary school, or special school [38]. Special needs experts can give recommendations, but the final decision concerning the choice of school is in the hands of the parents or carers [39]. In the academic year 2018–2019, 5% of students received special support in Austrian comprehensive schools. Of those, 63.1% studied full time in mainstream classes (integration/inclusive classes) and 36.9% studied in special schools [40].

In Finland, all students receive a 10-year comprehensive education (from age six to 16). There are still special classes that take place in mainstream schools and some special schools. The compulsory school system offers three types of educational support. The general support is meant for all students, and there is no need for official documentation [41]. To receive intensified support, a pedagogical assessment is required. This assessment is made by the teachers, the student, the parents, and the school welfare personnel who are dealing with the student and his/her family. The third tier of support, special support, requires an extensive assessment and a formal administrative decision. An individual educational plan (IEP) is made for all pupils within the special support [41]. In 2019, 7.4% of students received special support in Finnish comprehensive schools. Of those, 22.5% studied full time in mainstream classes, 43.7% studied partly in mainstream and partly in special education classes, and 34% studied full time in special education classes or in special schools [42].

In Lithuania, the idea of inclusive education is fairly new. Until the end of the Soviet regime in 1990, according to the Lithuanian state policy, students with SEN were kept separate from their peers in specialized boarding schools or stayed at home or in residential care homes [25]. Now, compulsory education starts at the age of six and ends at the age of 16. Nowadays there are pedagogical-psychological services (PPS) that support schools in Lithuania. Two forms of education for learners with special educational needs exist in general education schools: education in a general class, providing necessary student support; and education in a special class, usually for learners with intellectual disorders [43]. The schools aim to promote diversity in educational establishments by creating favorable learning conditions for all learners, according to their needs and abilities. The studies are designed in collaboration with the students and their guardians. The Child Welfare Committees coordinate the arrangements for educational assistance. The committees consist of a school leader, various specialists, teacher representatives, and representatives of learners' parents [44]. In the academic year 2018–2019, the proportion of pupils with SEN in mainstream schools was 11.6%. Of those, 45.5% were enrolled part-time in mainstream schools and part-time in special classes or special schools [45].

The Polish educational system has gone through profound changes since the collapse of the communist regime in 1989. Now full-time compulsory education begins at the age of six as pre-primary education and the school applies to pupils aged 7–15 years. Part-time compulsory education (the obligation to be in school) concerns pupils aged 15–18 [46]. Their education may take place either in school settings (upper secondary school) or in non-school settings (vocational training offered by employers). The education of students with special educational needs is provided in three main types of schools: mainstream (inclusive) schools, integrated schools, or special schools [47]. In integrated schools, which are not necessarily located in the area of residence of the children, students with SEN study with their nondisabled peers or in special classes purposefully created for their needs (e.g., specific learning disorders or language disorders) [46]. Inclusive schools are regional schools where students with SEN study together with their peers [47]. In the Polish public education system, every student has the right to voluntary and free

psychological and pedagogical assistance [48]. The curriculum, methods, and forms of work shall be adapted for the child/student on the basis of a developed individual educational and therapeutic program (IPET), considering the recommendations of the Special Education Needs Assessment [46]. Each school provides counselling and guidance to students, parents, and teachers, in line with individual needs. Such support is also offered by counselling and guidance centers (referred to as psychological and pedagogical centers) [46]. In the academic year 2018–2019, 3.8% of students in Polish compulsory education were identified having special educational needs. Of those, 31.1% studied in special schools and 66.6% studied in schools with a non-segregated system. The students with SEN in those schools were studying in integrated classes (29.6%), in mainstream (inclusive) classes (68.6%), or in special classes (1.8%) in public schools [49].

There are some similarities and differences in terms of the provision of educational support in Austria, Finland, Lithuania, and Poland. Specialist (psychologists, speech therapists, and the like) team support is ensured for students with diverse needs in all four European countries, as well as designing and following an Individual Learning Plan or Educational Plan/program. The teachers and special education teachers of Austria, Finland, Lithuania, and Poland provide educational support for students based on their needs. Teaching/educational assistants have their responsibilities in the system of educational support, too, whereas learning support teachers work in schools in Austria. School Welfare groups/boards contribute to the provision of educational support in Finland and Lithuania. The educational support situation in Poland differs from that in Finland, Austria, and Lithuania as it particularly focuses on the assessment of a student's level of functioning and special educational needs. Nevertheless, some steps are being taken in Poland to reorganize the general curriculum, and to adapt it to the education of students with diverse needs, not just students with special needs.

## 3. Materials and Methods

This research presents comparative educational research and especially micro-level comparisons [50]. Based on the categorization by Rust, Soumaré, Pescador, and Shibuya [51], this research belongs to field research studies in comparative education research, which is common in micro-level comparisons. This research strategy employs data collection methods typical of this type of comparative research (in this case, interviews and pedagogical reflective journals). Details of data collection in practice are given after introducing the cases.

The research design of the original research carried out in the four European countries was developed during systemic scientific colloquia involving researchers from all the universities in the four European countries participating in the project, as well as expert teachers working in participating schools [25]. The research was organized by applying an interactional ethnography perspective [52]. The researchers implemented a multidimensional research strategy that enabled them, through various research methods, to analyze the processes taking place in selected schools.

### 3.1. Research Participants

A group of researchers representing one teacher training university in each of the countries joined together for research on inclusive education. Each university selected one school implementing inclusive education in their country. The schools are represented in Table 2. The schools were selected for their broad experience of inclusive education and reputation in the pedagogical community.

**Table 2.** The university and the school communities in each country.

| The Country | The University | The School |
|---|---|---|
| Austria | University College of Teacher Education, Vienna | Integrated Learning Center Brigittenau, Vienna |
| Finland | University of Lapland, Rovaniemi | Teacher Training School of University of Lapland, Rovaniemi |
| Lithuania | Lithuanian University of Educational Sciences, Vilnius | Vilnius "Versmės" Catholic School, Vilnius |
| Poland | Pedagogical University of Cracow, Cracow | School No. 12 with Integrated Units, Krakow |

Two class communities, including teachers, pupils and their parents, from each school participated in the research (Table 3). All the classes were at an elementary level, the students' ages being seven to 12. All participants gave informed consent after reading relevant information about the research goals and the means of publication, as well being given the option to withdraw from the research whenever they wanted [53].

**Table 3.** Summary of the classes in the research.

| Participant | Finland | Lithuania | Poland | Austria |
|---|---|---|---|---|
| Students | 42 | 45 | 40 | 36 |
| Teachers | 2 | 2 | 2 | 2 |

However, our research results reflect the social pedagogical interaction in the entire community of the participating school (including school leaders, special pedagogical assistance providers, etc.), as well as national and local administrators outside of the school community.

*3.2. Research Instruments*

All four research partners implemented similar research methods when carrying out the original study. The original data included teacher, parent and student interviews, observation, sociograms and teachers' pedagogical journals. In addition, the data collection took place within the same time period in each country (spring and autumn 2015). This article is based on teachers' interviews and pedagogical journals, as these formed the primary data for the questions set for this research. The data and their reference codes are introduced in Table 4. The other data served as supplementary data in this article.

**Table 4.** Research data and the reference codes.

| Data | Extent of Data | Reference Code * |
|---|---|---|
| Teacher interviews | Three interviews with each teacher | Interview, T1 or T2 = Teacher 1 or 2 |
| Pedagogical journals | Each teacher wrote for one week, from September to December | Diary, TA or TB = Teacher A or B |

* Each nationality was added at the end of the code: AU—Austria; FI—Finland; LIT—Lithuania; POL—Poland, e.g., T2/LIT means Teacher 2 from Lithuania.

The interviews in every country included two class teachers or special education teachers from each participating school. All teachers were interviewed three times (April–November 2015). The main topics of the interviews were the following: (1) the relationships between the pupils in the class, (2) the relationships between the teachers and the pupils in the class and (3) the co-operation and communication between professionals in the school and the professional network. The interviews were semi-structured as the researchers planned together the main questions to be asked (see Appendix A). The length of the interviews varied between 17 to 58 min. The average length of the interviews was 30 min. All the interviews were transcribed. The researchers did not agree upon the font type but the minimum number of words in one interview was 2235 words (17 min interview) and the maximum number of words was 9561 words (in 58 min interview). The teachers kept pedagogical journals for four weeks, one week per month from September to December in 2015, in which they reflected on their experience of implementing inclusive education. They reflected on the experiences of identifying and acknowledging diverse needs, planning

curricula and activities, organizing support to students and teachers, and developing cooperative networks. The topics and instructive questions for the pedagogical journals are presented in Appendix B. The lengths of the pedagogical journals varied a lot depending on the teacher's way of expressing one self. The shortest journal was 5174 words and the longest was 21,533 words.

For the selection of analyzing the strategy applied in this research, it was important to acknowledge that, for "studies involving direct, concrete, and subjective experience with reality" [51], p. 104, the worldview the research presented was constructivist in nature [54]. The method of structuring content analysis [53,55] was applied for the analysis of qualitative data. The qualitative data from interviews and pedagogical journals were analyzed in the following stages: stage 1—definition of the categories; stage 2—description of examples in each category; stage 3—deductive analysis, producing text with references to the categories, moving from theory to the text, and stage 4—inductive analysis, moving from text to theory, open for new categories. The research results were then compared between the countries.

## 4. Results

As a theoretical framework, we used Rose and Norwich's [8] analysis of interprofessional work. We analyzed the findings on two levels: the social level, introducing the policy context; and the social-psychological level, describing the team members' experiences.

### 4.1. Introduction of the Four School Cases

The sociocultural context of the school cases differed considerably. Thus, the results section begins with a short introduction to each school case. The introduction of the school cases is meant to frame the policy context, i.e., the national regulations and school policies, and to deepen the analysis of the categories and the comparison of the school cases.

#### 4.1.1. The Integrated Learning Center Brigittenau, Vienna, Austria

Vienna, in contrast to many other parts of Austria, is a culturally diverse city [56]. In the Integrated Learning Center Brigittenau (ILB), all classes are inclusive, multi-age classes, each containing four to five pupils with special educational needs and students from an immigrant background. In the academic year 2014-2015 there were 368 students, of whom 91 were identified as having special educational needs [57].

The organization of teaching and learning is in three stages depending on the age of the pupils. The first to third are is called the entry stage. These pupils are taught in six inclusive classes of approximately 20 pupils. The fourth to sixth grades are the transition stage. The pupils are organized into inclusive clusters of approximately 44 pupils. The classrooms are designed as subject rooms for German language, English language, mathematics, or science. The seventh and eighth grades are the exit stage. In all of these inclusive groups, the children have individual plans for their learning. In the first to third grades, the pupils stay in their social group, but their work is also individualized. In the fourth to eighth grades, the pupils are given a topic or question and tasks from the subject teacher in the subject room. Music, art, handicraft, and gymnastic lessons are weekly, and fixed by the classroom teachers [58].

There are different types of teachers in every class: a special needs teacher and a class teacher or, for the older students, a subject teacher. Once a week for 2 h, teachers have team meetings after the school lessons. To organize the students' individualized work, there is a coaching system whereby each teacher takes care of one group. Approximately 10 pupils are in one coaching group. These groups meet regularly, most of them in the morning circles, to plan their work or individual projects and discuss further steps. There they also talk about other problems the pupils need help with. During the lessons, there are always at least two, and sometimes three or four, teachers in the classroom, and various forms of team teaching are applied. There are five tutoring lessons per week during which subject

material learned can be revised and memorized. Lessons are based on reform pedagogy, as in Montessori, Freinet or Jenaplan pedagogy [57].

Lessons start at 8:30. The first learning unit lasts for 100 min, followed by a 30-min break. Self-directed learning phases, such as independent work on individual projects, are discussed and structured together with the teachers. There are ordinary lessons in German, mathematics, and English, wherein new contents are introduced by subject teachers and then structured into work plans. This structure is only interrupted when the groups work on larger projects. In the projects, everyone can contribute with their individual skills, and everyone is important for the community. The second learning unit takes place after lunch, lasts for another 100–150 min, and follows a similar structure to that of the morning one. ILB is a full-time school. All students are at the school from 8.30 a.m. to 4 or 5 p.m. In the afternoons, there are leisure teachers who are also members of the pedagogical teams. Other experts, such as advisory teachers and speech therapists, come to the school for several hours a week [57].

There are class council meetings once a week where students are trained in problem-solving and participatory skills. For peer conflicts, peer mediation is used, and students can participate in peer mediation training if they wish. The forum for adult debate is the Class Parliament where the teachers and parents can have discussions [58]. Moreover, at the school level there is the School Parliament, where the teachers and parents have their representatives [59]. Once a week, parents meet in the "Parents' Café". This serves as an opportunity to receive information on school life in a rather informal setting and in languages other than German. These meetings are organized by an external association and are led by a multilingual organization. A strong parents' association [59] cooperates with the headmaster, the staff, and the management board in order to build effective cooperation and partnerships between the pupils' home and school lives. In various work groups, the parents, teachers, and pupils effectively deal with specific issues.

### 4.1.2. The Teacher Training School, Rovaniemi, Finland

In the Teacher Training School of the University of Lapland, the number of children was about 317 in the academic year 2014–2015. At the school, there were six grades, with three parallel classes in each grade, for a total of 18 groups. Every year, there are also about 200 student teachers practicing in the school; accredited teachers are responsible for supervising them.

In the Teacher Training School, teachers and teaching assistants have a weekly 1-h planning and reporting meeting. Moreover, teachers meet other professionals, such as therapists, school social workers, and family counsellors, when they have student welfare matters with their students. Parents' evenings take place several times a year. Teachers also offer family meetings when necessary, but at least once a year, usually for 1 h at the end of the school day or in the evening. Usually the student him/herself attends the meetings at least some of the time [60].

According to national practice, in every class, some pupils receive intensified support and a few receive special support, alongside the support and teaching that all pupils receive. In other words, in this school, all pupils with special needs study in their "home classes," with a special education teacher coming into the home classes as a co-teacher a few times a week. Every student with special needs also has the opportunity of learning part-time in a flexible small group [41].

At the beginning of the school year, class teachers and special needs teachers, in co-operation with the parents and the student, plan the special support for the students individually. Again, the plans are assessed at the end of the school year together with parents and the student. At the school, there are three special needs teachers, one of whom is in charge of those special needs children who would benefit from individual teaching and learning in a flexible small group. Similarly, there are three trained teaching assistants able to help in various classes, moving between them according to their needs [60].

The school carries out remedial work with students in groups of a maximum of 10 students. A group called "Maltti" (Patience) is one such remedial program, guiding children to self-determination [61]. Taught by special education teachers, it supports children who have difficulties in concentrating on learning and controlling their own feelings and behavior. Another remedial program is the so-called ART group (Aggression Replacement Training), run by the school social worker for antisocial and choleric children. For individual student welfare services there are several professionals available, organized by the municipal district—for example, support from the school social worker, school psychologist, the family counselling, and rehabilitation and children's psychiatric clinics [62], or regular support from therapists (e.g., physiotherapy, speech therapy, functional therapy, or riding therapy).

The teachers use various methods to help pupils concentrate, focus on learning, participate, and learn in their own ways and at their own levels. Special attention is given to various teaching methods, flexible grouping, anticipating and preparing pupils for future situations, and task differentiation. To support the schoolwork in the classrooms, teachers also use daily routines, structuring, and modelling. Importance is also given to visualizing, concretizing, chopping assignments into small parts, taking little breaks during lessons, exercising during the lesson and throughout the school day, and awarding systems. Teaching is based on flexible grouping and utilization of space, personal guidance, co-teaching, and part-time special education, and, in many cases, on multiprofessional work. [60].

### 4.1.3. Versmė Catholic School of Vilnius, Lithuania

After the signing of the Act on the Restoration of Lithuania's Independence on 11 March 1990, an active movement began developing a society and educational system based on democratic foundations [37]. The processes were stimulated by the initiatives of emerging nongovernmental organizations and community groups. One of these initiatives was realized by the teachers working in a boarding school for blind children. The Versmė Catholic School of Vilnius started inclusive education practices in Lithuania, with the help of a group of pupils' parents. The community of Versmė Catholic School of Vilnius sought to enable all students, including those with disabilities, to learn, play, and relax together. The community set itself another goal: to prove that boarding school is not a prerequisite for learning for students with disabilities. The socio-educational basis of the school was generated by analyzing the best practices of American teachers, and the school education model was supported by scientific research. The school ethos was developed on Christian values. The Versmė school soon became a model of inclusive education practice for other schools in Lithuania, and its leaders and teachers were active participants in shaping inclusive education policy in Lithuania.

In the academic year 2014–2015, there were 579 pupils in the school, 80 teachers and specialists (psychologists, special needs teachers, social workers, physiotherapists, and speech and language therapists), and 20 teaching assistants. There were two parallel classrooms of primary, basic, and secondary education programs in the school. Up to 30% of pupils in each classroom had disabilities, such as visual or hearing impairments, movement disorders, autism spectrum disorders, mental disabilities, etc. Children with severe mental disability were educated in a special class.

The educational system of the school aims to enable each pupil to achieve the highest personal academic results possible and to develop the capacity to live and be active in a heterogeneous community. The whole school environment is adapted for pupils with reduced mobility, so they can move independently using wheelchairs or special supports throughout the school environment. Teaching assistants and professionals support pupils with learning difficulties as well. The teachers, acting together with special education teachers, select the directions of educational differentiation, appropriate educational means, and ways of organizing activities. Support for teachers is provided using the resources

of the school and external institutions. Pupils' needs assessment and institutional and interinstitutional support are coordinated by the Child Welfare Committee [63].

The school fosters positive interactions between teachers and pupils, which helps to create a trust-based educational environment that ensures the well-being and safety of all students. One of the most important strategies in a heterogeneous school community is collaboration when identifying common goals, coordinating actions, and empowering each other. An important tool for enhancing positive interaction is the learning of communication skills [64]. In heterogeneous classrooms, teachers employ empowering education. As teaching methods go, teachers rely on, e.g., sharing information and experiences, searching for collective solutions, reflecting on results, promoting inspirational examples, and enhancing conscious commitment. These directions of education enable all pupils actively participate and express themselves in educational processes. In this way, teachers' support for pupils is focused on learning activities without excluding some of the pupils or overemphasizing the need for support. The practice of empowering education enables pupils with adapted programs to successfully engage and participate in the joint learning activities. In addition, teaching assistants and other support professionals are a natural part of the learning environment [65].

### 4.1.4. Primary School No. 12 with Integrated Units, Krakow, Poland

Primary School No. 12 is a public school (inclusive) with integrated units. This was the first school in Poland to include students with special educational needs. Since 1957, the school has operated as a teacher training school, providing practical training for higher education students. In 1999–2019, this institution functioned as the Zespół Szkół Ogólnokształcących Integracyjnych (Union of Integrated Schools) and consisted of six grades (Primary School No. 12) and a three-grade Gymnasium No. 15, with Integration Units. In each grade there are two to four peer classes, including one or two classes of an integrated type. In the academic year 2014–2015 a total of 610 pupils attended the school, including about 100 pupils with special educational needs due to various disabilities [66].

In the integrated classes there are 15 to 20 students, including 3–5 students with different types of disabilities. Each integrated class employs two teachers at the same time. At the level of the so-called junior classes (grades 1–3), where the research was carried out, there was one early years education teacher and a special education teacher. Both these educators worked full-time in the classroom, i.e., 20–23 h per week. Physical education, foreign languages, and computer science classes were conducted by an additional teacher, and students were also accompanied by a special education teacher.

It is the rule that, at the beginning of each school year, an individual educational and therapeutic program (Indywidualny Program Edukacyjno-Terapeutyczny, IPET) is prepared for each student with SEN. It provides information on the strengths and weaknesses of the pupil, possible curriculum modifications, individual activities (for each SEN student two additional hours are organized per week) and team activities to help each student succeed. This school employs many specialists, including numerous special education teachers, physiotherapists, speech therapists, psychologists, and social skills training specialists. In addition, there is much systematic cooperation between teachers and parents, who are treated as partners in their children's education [67].

There are 22 classrooms available to pupils and two gym halls, one of which is used for choreotherapy, as well as a biofeedback laboratory, speech therapy rooms, and a correctional and compensatory room for all students who need support. The school is fully adapted to work with pupils with special educational needs, not only because of the universal local layout, which eliminates all physical barriers, whether architectural or spatial, or those related to the environment, but also because of its organization, which promotes integration [66].

The school is an open and dynamically developing institution focused on the changing needs of students, the expectations of parents, and the possibilities of teachers. The school aims to ensure the comprehensive and harmonious development of the child both

individually and socially as a member of the collective (class, school, local community, nation, European community, and—most broadly—citizen of the modern world).

The school's mission is to continuously support each pupil on the path to broadly understood success—his/her harmonious, comprehensive development, in accordance with his/her personal needs and abilities. The school sets itself the goal that the spontaneous cognitive motivation of the child, as a result of education and interactions, will develop into conscious motivation [68].

*4.2. Comparison of the Policy Context between the School Cases*

For the comparison of the policy context, we used the elements of Rose and Norwich's [8] social-level analysis, highlighting national regulations and school policies. The school cases were quite similar in terms of the multiprofessional support system. In some form, they all offered multiprofessional support for teachers and their students and families. Our data did not allow us to more deeply analyze the possible interactions and tensions between different policies that may occur when applying the municipal regulations at the school level [8].

However, the data and the information of the national educational regulations show that the school cases differ in terms of their pedagogical structures. The strongest pedagogical support, enhancing inclusive environments, existed in the Austrian Brigittenau and in the Polish Primary School No. 12 school cases. As a basic assumption, both these schools were built on teachers' co-teaching. In Brigittenau, all classes had at least two teachers, one class or subject teacher, and one special education teacher. In the Polish Primary School No. 12, there was the same kind of teaching resource in the integrated classes. In the Lithuanian and Finnish schools, there was a collaboration-based support structure, too, but there were fewer teachers available or the special education teachers were responsible for several classes. In addition, all of the schools had spaces for various kinds of functions, not only traditional classrooms, but, e.g., for rehabilitative action.

At all of the schools, support was made available in the students' daily lives. Many kinds of consultation were available and other professionals supported the students at school, e.g., giving occupational therapy, school social workers participating in lessons, helping teachers to develop the social relationships in the class, and psychological services. However, in Poland, the curriculum and guidelines for educational programs were quite detailed and regulated, whereas, according to Austrian, Finnish, and Lithuanian regulations, the national guidelines gave more flexible possibilities for teachers and other professionals to shape learning environments to match the whole student group's diverse needs.

*4.3. Analysis of the Social-Psychological Level of Interprofessional Work and Its Dilemmas*

Our ethnographic data gave us opportunities to look at the social-psychological level of interprofessional work. We categorized the data according to Rose and Norwich's [8] theoretical framework, which, for a social-psychological-level analysis, has three categories, collective commitment, efficacy, and process. Within these categories, we also paid attention to possible dilemmas, but, since our original research aim was to explore good ways of implementing inclusive education, there are few references to dilemmas in the data.

### 4.3.1. Collaborative Commitment

In all datasets, the teachers' work was based on shared values and insights that made it possible for them to find mutual goals when working with each other, other professionals, as well as with students' parents and carers. Through this, they were collectively committed to working together and saw themselves as responsible for ensuring positive outcomes [8].

> "I cooperate with a wide and meaningful network: parents, colleagues (especially those who teach in the same grade and special education teachers), school counsellors, trainees, speech therapists, occupational therapists, and physiotherapists . . . school nurses, and school social workers. Within the network, we share

news, know-how, and we consider possible forms of support for the school's everyday life."

(Ped. journal, T1, FI)

The strongest ties seemed to exist among the Lithuanian Versmė school members. They also spent time together during their leisure time and often had informal get-togethers: "Before the school year, in December before Christmas Eve, and at the end of the school year we go on retreats" (Interview, T1, LIT). By contrast, the Austrian data showed that the teachers preferred to maintain correct and professional relationships with their colleagues (Interview T1, T2, AU).

All of the teachers involved in our research also saw boundary spaces [23] in their collaboration, which means they saw opportunities to complement each other's skills and thus obtain benefits for their own work among the students as well as promote the goals they set for their work. This required coordinated action and the division of work tasks in order to achieve a joint goal [8].

> "From a professional perspective, one could say that each person's skills and competence are included in the teaching process, which can be quite fruitful for the team."
>
> (Interview, T1, AU)
>
> "I, as a teacher with experience in working with pupils with disabilities, know how important it is to support the young as well as those who have been teaching a single subject in the classes where the pupils were rather average."
>
> (Ped. journal, T1, POL)

Teachers characterized the work in boundary spaces as requiring recognition of other members' professionalism, as well as respect for opposing opinions and the ability to compromise.

> "I believe it is a constant, almost daily struggle to find similarities, to distance yourself from certain things, to give in, and to have your way as well."
>
> (Interview T1, AU)

### 4.3.2. Collaborative Efficacy

The second category refers to the group's beliefs about its capabilities to carry out a course of action [8]. Efficacy here means both individual as well as interactive aspects of group functioning as a whole [8]. The teachers in our research were aware that working together needs supportive interaction as well as respect for other members' knowledge and skills. The teachers utilized consultations by other professionals, a service that was regularly available in schools.

> "If they need professional consulting and coaching, professionals can come to school and the teachers can discuss problems and conflicts in professional settings. In one school year I and my team arranged six consulting sessions."
>
> (Interview T1, AU)
>
> "Solutions to the psychological issues of these girls were discussed with the psychologist more than once—The psychologist reassured me by saying that the children will only learn through consequences."
>
> (Ped. journal, T1, LIT)

In addition, teachers appreciated consulting and discussing with colleagues, as they could share good practices and discuss current issues with their teaching. This was apparent especially from the Lithuanian and Polish data.

> "We have methodological meetings where we discuss educational matters and new things that we have learnt. The meetings are pre-planned because it is always interesting to know more."

(Interview, T2, LIT)

"We also have teams, and each time a need arises, we meet; we get on well. We must find an adequate solution if something is happening in the class; we act together."

(Ped. journal, T2, POL)

The research data offered several indications that teachers felt that their professional competencies complemented each other. For example, Finnish teacher 2 described her collaboration with the special education teacher:

"Our collaboration has its roots way back. We both know what we like, and we try to bring those elements to the co-teaching lessons. We have a clear [division of labor] and she knows my system in the class."

(Interview, T2, FI)

4.3.3. Collaborative Process

The third category describes the ways in which the teachers planned and implemented their teaching and collaboration with others. In the school cases, reflection was one of the most powerful ways to understand the changing situation at a school. The constant reflection enabled flexible implementation of teaching arrangements and construction of student support [10].

For example, in the Finnish school case, teacher 2 preferred co-teaching but this year, she and her colleagues had to develop flexible hybrid models for co-teaching according to this year's students' needs:

"This year, we [the class teacher, the special education teacher, and the resource teacher] have often divided the group flexibly so that the one group has stayed at our classroom, and the other group has worked the next door, at least for part of the lesson. There are several pupils for whom mindfulness and concentration are challenging."

(Ped. journal, T2, FI)

In spite of the good practices and successful execution of collaborative work, the teachers identified some dilemmas that complicated their work. Partly, the difficulties were connected to too few resources in terms of time or personnel. For example, in the Austrian school case, teacher 1 felt that the teachers were obliged to concentrate too much on organizational matters in their weekly 2-h meetings, instead of having more time for pedagogical exchange (Interview, T1, AU). The same notion came up in the Finnish teachers' data (Interview T1, T2, FI).

There were also instances where the teachers' collaboration did not work out well. It seemed that some teachers were not committed to inclusive goals or wanted to keep their traditional autonomous role as a teacher.

"Some teachers' have a positive attitude towards co-teaching and they enjoy it. Some teachers, on the other hand, find it stressful. Some think that co-teaching is only a passing phase because it is too expensive anyhow."

(Ped. journal, T2, FI)

In the Austrian school case, teacher 1 described a poorly planned lesson where the responsibilities were not clear and the pupils became restless, chatting loudly. In addition, the teacher felt she did not receive support from her colleagues (Ped. journal, T1, AU). This occasion is linked to teachers' professional identity or the dilemmas in controlling the teaching situations collaboratively, supporting each other instead of holding on to the traditional teacher's role.

In sum, our data showed that team members' ability to have conscious dialogical conversations with each other is crucial to the success of interprofessional work.

> "I help my colleagues, they help me; for instance, when Jaś had communication and aggression problems, we all sat together and each of us tried to suggest some solutions. It did not end with conversation only; we acted together and it turned out that after a few such meetings everything started to go well somehow."
> (Ped. journal, T1, POL)

The teachers in our research pointed out that precise and joint planning and mutual understanding form the backbone of successful inclusive teaching. They also accentuated the importance of negotiation and a division of responsibilities and roles (e.g., Interview, T1, AU and Interview T2, FI).

## 5. Discussion

Based on our results, collaborative action seemed to be an essential part of teachers' work in an inclusive school. In our school cases, the supportive collaborative actions were characterized by two main features, proactive and reactive ways of constructing an inclusive pedagogy. As proactive structures, we see the kinds of solutions that benefit all the students [5]. For example, when co-teaching is the foundation of supporting students, teachers can build learning environments that consider diverse students' needs.

Another dimension of proactive structures is collective commitment. A high level of commitment characterizes an inclusive school community, where all personnel are supposed to work together in line with collective goals [6,8]. In our school cases' data, there were quite clear manifestations of collective commitment to inclusive values, such as valuing every student and family and seeing every student as a transformative learner [5]. Still, in the Finnish school case, the data showed that some of the teachers at the school had diverse ideas when it comes to inclusive goals. This is probably due to the autonomous position of Finnish teachers and the different generations of teachers; when they have acquired their education may affect the professional values and role they have adopted [9]. While the Finnish school had an average level of commitment to inclusion, the other three school cases were exceptional in terms of acting as pioneer schools in the field of inclusion in their own countries.

The other main feature of teachers' collaborative work is reactive structures. Here, we include the principle that students' difficulties with learning can be considered dilemmas for teaching as a whole rather than problems with the students themselves [5]. This notion acts as the basis for considering what resources are sufficient when it comes to the number of guiding adults in the learning community, what kinds of teaching methods are carried out, what the division of work is among the teachers and other professionals, and how and by whom the students' well-being is supported.

Of course, it is worth keeping in mind that the data acquired in this research were from four schools, each one representing a country. Thus, the work culture in these specific schools has likely affected the findings, in addition to wider cultural factors and national guidelines for education. On the other hand, the purpose of this research was not so much to introduce generalizable findings but to show through case studies how inclusive teaching practices were implemented and could be developed based on experience and observation. The data included various sets that complemented each other and therefore increased the reliability of the research. The differences between cultures and countries provided perspective on inclusion and the ways and levels of its implementation, so that through the analysis of these examples, it can be further developed worldwide.

When personnel work collaboratively towards inclusive goals, they need time and space to reflect on mutual values, conceptualize the polices they aspire to, and renew professional knowledge and skills. All this means, as Tynjälä [69] states in her study of work place learning, integrating theoretical, practical, and self-regulative knowledge in formal and informal situations. In their comparative research of interprofessional and collaborative work in health, social care, and education, Floyd and Morrison [13] noted the extreme complexity of the abovementioned concepts. The researchers accentuate, as did Rose and Norwich [8], that, at the macro level, the concepts need to be reflected with

existing regulations, policies and practices, and in micro level as personal biographies and abilities to amend one's professional identity, role, and control.

## 6. Conclusions

Our results support the conceptual framework of inter-professional teamwork created by Rose and Norwich [8]. Interprofessional and collaborative work appear as transforming and negotiating processes. Our research results indicate that collaborative actions need to be an essential part of teachers' work in an inclusive school. Partly, the heterogeneous students' needs can be anticipated and their skills can be enhanced by collaborative proactive ways of constructing practices. Partly, reactive ways of constructing practices are needed, where teachers and other professionals together reflect their mutual goals and own actions in order to build socially sustainable learning environments and social arenas for their students and their families. As such, our results could be implemented, for example, in teacher education, indicating the importance of developing student teachers' reflective and dialogical skills and awareness of collaborative ways of working. Another implication could be strengthening the previously mentioned skills through teachers' and principals' in-service training.

**Author Contributions:** S.L. was the main author of this article and performed the theoretical review and main analyses, and outlined the conclusions. A.G. coordinated the design of the data collection in the original research; J.N., T.C., S.T., S.L., and A.G. gathered the data and did the initial analysis. All the authors acted as experts on their own country's educational system. S.U. contributed to the theoretical review, methodological writing, and conclusions. All authors have read and agreed to the published version of the manuscript.

**Funding:** This study received funding via the European Commission Erasmus+ project "Inclusive Education: Socio-Psychological, Educational and Social Aspects" (No. 2014-1-FI01-KA200-000893).

**Institutional Review Board Statement:** Ethical review and approval of an institutional board were waived for this study, due to reason that the research procedures and practices followed All European Federation of Academies of Sciences and Humanities'instructions.

**Informed Consent Statement:** Informed consent was obtained from all subjects involved in the study.

**Acknowledgments:** We wish to thank the other members of our original international researcher team, Sabine Albert, Georg Jäggle, Agnė Juškevičienė, Remigiusz Kijak, Outi Kyrö-Ämmälä, Joanna Kossewska, Ona Monkevičienė, Katja Norvapalo, and Stasė Ustilaitė, who gathered all the research data. We are grateful to the teachers, students, and parents who participated in our research and made their experiential expertise available to us.

**Conflicts of Interest:** The authors declare no conflict of interest. The funders had no role in the design of the study; in the collection, analyses, or interpretation of data; in the writing of the manuscript, or in the decision to publish the results.

## Appendix A

**Table A1.** Interview frame for teachers.

| Relationships among pupils | Time period |
| --- | --- |
| Please share about the interpersonal relationships among pupils in your classroom:<br>1. What signs of equality, respect, and dignity do you observe?<br>2. What signs of acceptance, pleasure to be together, and tolerance do you observe?<br>3. What signs of support, and cooperation do you observe?<br>4. How do you create the relationships in the classroom?<br>5. How do you try to solve relationship-related problems in the classroom? | April, 2015 |

**Table A1.** *Cont.*

| Teacher's relationships with pupils | |
|---|---|
| 1. Please share on what makes you happy in your relationships with your pupils? 2. How can pupils participate in everyday decision-making process in your classroom? Please provide specific examples. 3. What possibilities do you have to talk, to support, to motivate, and to encourage pupils individually? 4. What do the pupils want to talk to you about the most? Please provide specific examples. 5. How can you recognise the different needs of pupils, and how do you create positive relationships with them? 6. What would you like to change in your relationships with the pupils? | April, 2015 |
| School community | |
| 1. Please share the ways you create ethos in your heterogeneous school community. Do teachers have common leisure time? Please provide specific examples. 2. Please share on how you create professional relationships in the community of teachers? 3. How do the teachers share the good and bad experiences? Please provide an example of something valuable you have learned from your colleagues. 4. What would you like to change in the relationships within the community of teachers, why and how? | October– November, 2015 |

**Appendix B**

**Table A2.** Teachers' pedagogical journal, instructions.

| The field of education | Basic questions |
|---|---|
| Design of learning Teacher as a support seeker Teacher as a support giver Cooperation and my experience in it Meeting the needs of learners: planning, implementing, evaluating Reflection and evaluation of my teaching process Learning environment: What would I have needed more of today? Learning environment: What did I appreciate today? | How do I do it? What challenges do I meet? How do I/we solve the problems in certain circumstances? How do I/we overcome it? What was there of the best today? What was there of the best this week? |

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
