# Peer review of "Teachers Supporting Students in Collaborative Ways—An Analysis of Collaborative Work Creating Supportive Learning Environments for Every Student in a School: Cases from Austria, Finland, Lithuania, and Poland"

_sustainability, doi:10.3390/su13052804_

Round 1

Reviewer 1 Report

The article presents the results of 4 case studies that analyze the culture of teacher collaboration in 4 schools in Austria, Finland, Lithuania and Poland.

It is a well-prepared work that represents an added value in the field of inclusion given the qualitative and international nature of the results it provides.

There are some issues that I think the authors should take into account and that I think can help improve some specific aspects of it:

1. Bibliography.
In general, the bibliography used to give theoretical support to the arguments presented is of quality and is correctly focused on inclusive education.
However, most of the references are relatively old. It is recommended to carry out a new bibliographic search that allows updating the references with the latest contributions in the area.

2. Objectives.
The objectives of the article appear several times throughout the article: at the beginning of page 3 "we are interested in the ways in which ..."; at the beginning of the Materials and Methods section "The purpose was to ana-lyze teachers’ multiprofessional ... "; "One of the main purposes is to identify the features ...".
The phrases in which these objectives are stated do not fully coincide in all the occasions in which they appear.
It is recommended to review this issue and state the objectives only once (preferably at the end of the instruction) in the clearest and most specific way possible.

3. Headings.
It is recommended to include a new sub-heading referring to instruments at the bottom of page 7, just before the beginning of the last paragraph.
In the current version of the article, the description of the instruments is under the heading "Research Participants".

4. Instruments.
One of the most relevant issues is that the research instruments used should be better described.
4.1. Interviews
Their duration should be specified, if they were recorded and if they were structured, semi-structured or open interviews should be informed. If there was a previous questionnaire, it could be added as an annex.
Likewise, it should be specified why three interviews were conducted with each teacher, how much time elapsed between one interview and another.

4.2. Pedagogical journals.
More information on these journals should be given. What instructions were given to teachers to prepare their journals? were they offered any specific training? How did you ensure that journals were produced in the 4 countries in a reasonably similar way?

4.3. Observation.
Were there previous observation records? Or were the observed situations directly noted in a white paper?
Could this procedure be qualified as participant observation?

4.4. Sociograms.
Table 4 lists the sociograms, but no information about them is provided in the results.
It is suggested either to remove the information from the sociograms; or to expose their results.
Given the length of the article, it is suggested not to extend it further.

5. Structure of the article.
It is suggested to relocate section 4.1 in the Materials and Methods section, since it deals with the description of the schools, which is not clear that it implies a result in itself.

6. Formal issues.
6.1. Keywords must be in alphabetic order.
6.2. The meaning of the acronym "PIT" must be specified the first time it appears (at the beginning of page 3).

Author Response

Dear reviewer,

attached you will find our revised manuscript. The changes are tracked. As we wrote the new version, we considered the your constructive comments, thank you for those! We have made the revisions that we saw important. Attached you will find the answers to your comments.

Suvi Lakkala

Reviewer 2 Report

This article focuses on an important topic. The idea and the design of the research are very clever. However, it would be good if the conclusions of the article could be deepened and presented in a separate chapter at the end, especially with regard to the question: What are the consequences of the results?

In section 2.1, when it refers to the Profile of Inclusive Teachers, the acronym PIT should appear immediately, so that it can be understood on the successive occasions when it appears.

In several paragraphs, references such as [see, e.g., 2, 3] [see also 55] [cf. 70Rust]. have been used which focus on previous studies or readings, in general, however, no further reference is made to the specific aspect referred to. The document would greatly benefit from additional information on each of the studies mentioned.

Finally, in the last paragraph, it refers to the fact that "Our results support the previous research outcomes". which, on the one hand, are not cited in the text and, on the other, it would improve the presentation and understanding of both the research presented and its conclusions, if a brief reference to these works could contextualise and strengthen its conclusions

Author Response

(The authors gave the same response as above.)
